# Perceived Effects of Orthognathic Surgery versus Orthodontic Camouflage Treatment of Convex Facial Profile Patients

**DOI:** 10.3390/jcm13010091

**Published:** 2023-12-23

**Authors:** Simos Psomiadis, Nikolaos Gkantidis, Iosif Sifakakis, Ioannis Iatrou

**Affiliations:** 1Department of Oral and Maxillofacial Surgery, School of Dentistry, National and Kapodistrian University of Athens, GR-11527 Athens, Greece; iiatrou.dent@gmail.com; 2Department of Orthodontics and Dentofacial Orthopedics, School of Dental Medicine, University of Bern, CH-3010 Bern, Switzerland; nikolaos.gkantidis@unibe.ch; 3Department of Orthodontics, School of Dentistry, National and Kapodistrian University of Athens, GR-11527 Athens, Greece; isifak@dent.uoa.gr

**Keywords:** patient outcome assessment, facial appearance, convex profile, dental overjet, orthodontics, orthognathic surgery

## Abstract

Increased facial profile convexity has a common occurrence in the population and is a primary reason for seeking orthodontic treatment. The present study aimed to compare the perceived changes in facial profile appearance between patients treated with combined orthognathic/orthodontic treatment versus only orthodontic camouflage treatment. For this reason, 18 pairs of before- and after-treatment facial profile photos per treatment group (n = 36 patients) were presented to four types of assessors (surgeons, orthodontists, patients, laypeople). Ratings were recorded on 100 mm visual analogue scales depicted in previously validated questionnaires. All rater groups identified minor positive changes in the facial profile appearance after exclusively orthodontic treatment, in contrast to substantial positive changes (14% to 18%) following combined orthodontic and orthognathic surgery. The differences between the two treatment approaches were slightly larger in the lower face and the chin than in the lips. The combined orthodontic and orthognathic surgery interventions were efficient in improving the facial appearance of patients with convex profile, whereas orthodontic treatment alone was not. Given the significant influence of facial aesthetics on various life aspects and its pivotal role in treatment demand and patient satisfaction, healthcare providers should take these findings into account when consulting adult patients with a convex facial profile.

## 1. Introduction

The skeletal and dental discrepancies characterized by a convex profile and increased dental overjet are common in the general population and within patients seeking combined orthodontic/orthognathic interventions [1]. There are three main treatment strategies to address such disharmonies. In growing patients, the enhancement of mandibular growth, sometimes combined with the restriction of maxillary growth, is a common approach, although its effectiveness in altering facial morphology has been questioned by recent evidence [2,3,4,5,6]. In individuals with no active growth, the option of orthodontic camouflage aims primarily in improving the dental occlusion and appearance whereas the more invasive, combined orthodontic/orthognathic surgery treatment aims not only in correcting a malocclusion, but in improving the facial appearance as well [7,8].

Convex profile patients with Angle Class II Division 1 malocclusion may have functional problems, such as traumatic occlusion or reduced masticatory efficiency [9], but the main reason for seeking orthodontic treatment is the improvement of facial appearance [10]. Considering patient satisfaction, the success of orthodontic treatment with or without additional orthognathic treatment is primarily determined by the perceived improvement of facial appearance [11,12]. An orthognathic surgery intervention is a relatively invasive treatment, which should meet the goal of an esthetic improvement of facial appearance, and thus, the primary patient expectation, apart from proper function [8].

Several previous studies aimed to quantify the effects of various treatment approaches on facial morphology, but how these are perceived by people determines the actual impact of an interventions on patients’ lives [7,13,14,15]. Therefore, the assessment of treatment outcomes, as perceived by different groups of raters, is fundamental for the application of evidence-based interventions. Patients treated with surgical mandibular advancement have reported greater satisfaction from their dentofacial images compared to those treated with orthodontic camouflage, with both groups reporting high overall satisfaction with treatment outcomes [16]. The amount of initial discrepancy has been emphasized as crucial for consistently perceiving improvement after surgery and for minimizing the incidence of profile worsening after treatment [17,18]. Other reports suggest the orthodontic camouflage as a viable treatment option, especially in the patients with full upper lips and a small mandibular deficiency [15,19]. On the contrary, there are reports in the literature suggesting that adult patients can be dissatisfied from the esthetic outcome of an orthodontic camouflage treatment [14,20,21]. Profile image simulation studies have shown favorable effects of both surgical and camouflage approaches, with surgical ones receiving higher scores from orthodontists and laypeople [22]. Other simulation studies emphasized the role of the nose in the perceived outcomes and favored more clearly the surgical approaches over camouflage treatment [23]. So far, there are very few studies that used actual patient images to compare the effects of orthodontic camouflage with surgical-orthodontic treatment and these present conflicting outcomes [19,21,24].

Based on the aforementioned arguments, it is clear that the classical clinical dilemma of orthodontic camouflage versus orthognathic surgery for treating a convex profile [25,26] remains a common challenge in contemporary practice. Therefore, the aim of the present study was to assess the treatment outcomes of combined orthognathic/orthodontic treatment versus orthodontic camouflage treatment in terms of the perceived change in facial profile appearance. This was performed through the presentation of actual pre- and post-treatment profile facial photos to various groups of assessors. The research hypothesis was that the perceived changes in facial profile appearance are not affected by the type of intervention, i.e., only orthodontic (camouflage) versus orthognathic/orthodontic treatment.

## 2. Materials and Methods

### 2.1. Ethical Approval

The study was conducted in accordance with the Declaration of Helsinki, and the protocol was approved by the Research Ethics Committee of the Dental School, National and Kapodistrian University of Athens, Greece prior to study commencement (date of approval: 22 June 2018, Protocol Number: 361). A written informed consent form was provided by all participants prior to the use of their data for research purposes.

### 2.2. Sample

The sample of the present study consisted of consecutively treated patients from the records of the Postgraduate Clinic of the Department of Orthodontics, Dental School, National and Kapodistrian University of Athens, Greece. The most recently treated patients with a convex facial profile and Class II Division 1 malocclusion were checked for eligibility aiming to create two groups of 18 patients (Group A and Group B), with a similar sex distribution. The determination of the sample size was based on empirical evidence, resource availability, and study feasibility considering the number of potentially eligible patients, as well as the number of raters. This sample size is considered appropriate for the tested outcomes [4,5,27]. Group A consisted of non-growing, convex profile patients who were treated with fixed orthodontic appliances in both jaws combined with orthognathic surgery in one or both jaws. Group B consisted of non-growing, convex profile patients who were treated only with fixed orthodontic appliances. The following eligibility criteria were applied in both groups:-Complete initial and final diagnostic data, i.e., history (medical, dental, and orthodontic/orthognathic treatment), initial panoramic and cephalometric radiographs, initial and final dental casts, initial and final intraoral and facial photographs (profile and frontal) of acceptable diagnostic quality.-Class II Division 1 dental anomaly at the start of treatment (bilateral molar Class II more than half cusp and overjet between 6 and 12 mm) with no considerable functional shift during maximum intercuspation (≥2 mm).-Convex skeletal configuration at treatment start (5° < ANB < 9° on the lateral cephalometric radiograph).-Convex facial profile at treatment start in the initial facial profile photograph (males: 15° < facial contour angle < 25°, females: 17° < facial contour angle < 27°) (Figure 1) [8].-FMA angle between 17.5° and 32.5° on the initial lateral cephalometric radiograph.-Total treatment duration of 1 to 5 years.-No rhinoplasty or other esthetic surgical intervention (including Botox treatment) on the facial soft tissues.-White European ancestry. This ancestry was largely overrepresented in the searched archives, as well as in the rater population, and, thus, we limited our sample accordingly to avoid confounding that could not be controlled.-Patients without congenital craniofacial anomalies, syndromes, marked facial asymmetries, and marked functional deviation during mouth closure (visual inspection by two authors independently).-Complete skeletal growth at treatment start (CS5–CS6 and age > 15 years).-Complete dental arch without missing teeth (excluding third molars) at treatment start.-Completed treatment (no discontinuation).-No treatment with fixed mandibular advancement devices (e.g., Herbst, Jasperjumper, Forsus).

At the sample selection stage, the initial diagnostic data were used, whereas the final diagnostic data were only checked for availability.

The pre- and post-treatment facial profile photographs of each patient were used to assess the perceived changes in facial appearance. These were taken with the Frankfurt–Horizontal plane parallel to the ground, the teeth in maximum intercuspation (light contact), and the lips in resting position, according to the clinic protocol.

**Figure 1 jcm-13-00091-f001:**
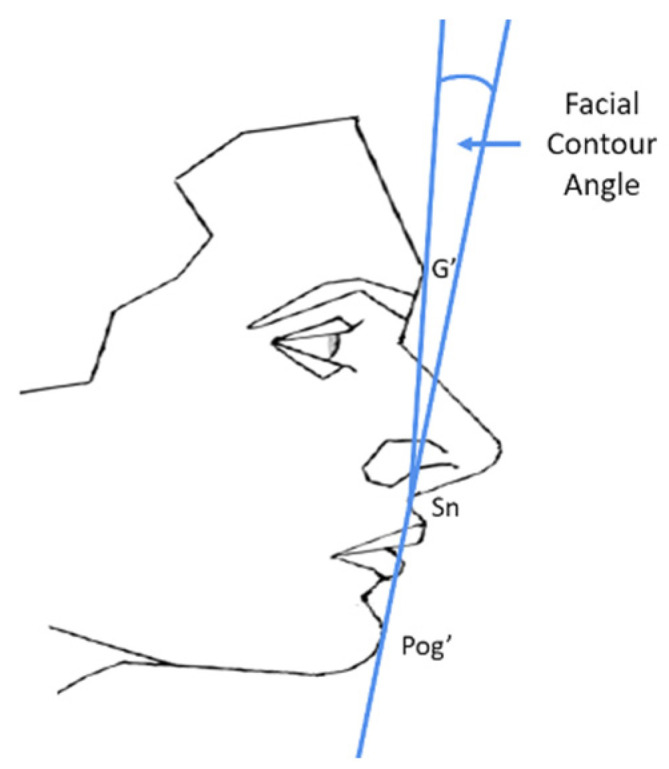
Drawing illustrating the facial contour angle, defined by the points: soft-tissue Glabella (G’), Subnasale (Sn), and soft-tissue Pogonion (Pog’).

### 2.3. Processing of Facial Profile Photographs

All digital photographs were edited in Adobe Photoshop (Version 22.0.1, Adobe Systems) to have a white background, similar brightness and contrast, and faces of equal vertical height (Na’ to Me’ soft tissue points). The photographs were assessed visually by three authors, independently, to identify any strong marks (moles, scars, etc.) or wiring elements (earrings, tattoos, etc.) that could influence the assessments, and these were digitally removed.

The adjusted pre- and post-treatment profile images of each patient were placed in a landscape-oriented A4 size (Figure 2) to be presented to the raters.

### 2.4. Rater Group Formation

Four groups of raters assessed the sets of patient images: (a) orthodontists, (b) oral and maxillofacial surgeons, (c) convex profile patients, and (d) laypeople. Each rater assessed 12 patients in one session to have standardized conditions while avoiding fatigue [4,5,27]. For this reason, the patient sample was randomly divided (https://www.random.org/, accessed on 23 June 2021) into three sets of twelve (six patients from each treatment group, with balanced sex distribution). In total, the patient photos were evaluated by 30 orthodontists, 30 oral and maxillofacial surgeons, 30 individuals with convex profile, and 60 laypersons. This ensured that 10 members from each of the first three groups and 20 laypersons assessed the photos of each individual patient. An increased number of raters was selected for the laypeople group to account for the diverse backgrounds of this group.

The rater groups of convex profile patients and laypeople had balanced sex distribution and consisted of white European subjects with variable educational level and socioeconomic status. Laypeople’s age ranged from 15 to 65 years. The convex profile patients were selected from the waiting room of the orthodontic department’s clinic. Care was taken to match their age and sex with the post-treatment age (±3 years or 1 year for individuals younger than 19 years of age) and sex of the rated patients. For the specialists’ groups, the first 30 specialized and final year resident physicians who agreed to participate were included. None of the raters were related to the studied patients in any way.

### 2.5. Questionnaire Fulfillment

Initially, each rater filled in a short personal detail questionnaire including name, age, gender, profession, and level of education.

Afterwards, the initial and final facial photographs of each patient were presented one by one to the raters for assessment (Figure 2). Half of the cases in each group (three boys and three girls) were randomly selected to be presented with their initial photos on the right and the final photos on the left, while the remaining were presented in an opposite way.

Each set of photographs was accompanied by a printed questionnaire, which was previously validated [4,5], and included five questions on a 100 mm visual analogue scale (VAS) to record ratings. Each of the questions was accompanied by a schematic illustration for easier understanding. The raters were asked to assess the changes in facial appearance that were evident between the left and right photos, on a scale ranging from “extremely negative” to “extremely positive” (Figure 3).

Two calibrated researchers administered all questionnaires in the same way and a pilot rating of a non-sample case was carried out. The raters were unaware of the purpose of the study and that the photographs illustrated treated patients. All questionnaires were completed in a quiet room, with adequate lighting, and under the discreet supervision of the researcher.

### 2.6. Recording of Measurements and Reliability Check

The distance from the left end of the VAS to each rater’s mark was measured using an electronic digital caliper (Jainmed, Seoul, Republic of Korea) to convert ratings to continuous variables. Data were recorded in millimeters, with two decimals, in a Microsoft Excel spreadsheet (Microsoft Corporation, Redmond, WA, USA). In cases where the final photographs were presented on the left page size, the VAS measurements were converted by subtracting the measured value from 100 to be analogous to the opposite ratings.

Thirty randomly selected responses were measured by the same researcher 2 weeks after the first measurement to assess method error.

Concerning the questionnaire validity, the intra-rater reliability has been tested previously for the same questionnaire and similar sample and rater populations and was satisfactory [4]. The validity of the questionnaire has been also tested thoroughly at various levels and was verified [4,5].

### 2.7. Statistical Analysis

The statistical analysis was conducted using the IBM SPSS statistics for Windows (Version 29.0. Armonk, NY, USA: IBM Corp). The homogeneity of variances was tested though Levene’s test and the data normality through the Shapiro–Wilk test and the visualization of relevant Q–Q plots and histograms. Parametric and non-parametric statistics were applied accordingly.

Group similarity in key characteristics was tested through a Mann–Whitney U test.

The systematic error of the repeatability in recording VAS ratings as metric variables was tested though paired *t*-test between 30 repeated recordings and the random error through the average and the standard deviation, as well as the range, of the differences.

Each patient was rated by 10 members of each rater group and by 20 laypeople. The median of these 10 ratings was considered a reliable assessment of a rater group and comprised the data used for further analysis.

The agreement between rater groups was tested through the intraclass correlation coefficient (ICC; two-way mixed model; absolute agreement; average measures). Values above 0.7 were considered as strong agreement, whereas values between 0.5 and 0.7 indicated moderate agreement. This analysis, along with comparative statistics between rater groups, denoted the concurrent and statistical conclusion validity of the questionnaires.

Differences between the orthognathic surgery and the camouflage orthodontic treatment groups were tested through two-way MANOVA. The responses of the raters to the five questions of the questionnaire were the dependent variables and the two treatment groups, as well as the four rater groups, were the independent variables. In case of significant results, various ANOVAs would be performed for each dependent variable separately and post-hoc analysis through Fisher’s least significant difference (LSD) test.

In all cases, a two-sided significance test was performed at an alpha level of 0.05. A Bonferroni correction was applied at this level, in the case of pairwise a posteriori multiple comparison tests.

## 3. Results

### 3.1. Method Error

There was no systematic error in the VAS rating measurements (paired *t*-test, *p* = 0.931) and the random error was also negligible (mean error: 0.00 ± 0.15 mm, range: −0.24, 0.27 mm).

### 3.2. Treatment Group Similarity

The treatment groups had equal sex distribution and similar age, treatment duration, facial convexity and overjet, before, as well after, treatment. The only difference between groups was in the change of the facial contour angle, with the orthognathic surgery patients presenting a significantly greater reduction in facial convexity than orthodontic patients (Table 1).

### 3.3. Perceived Facial Profile Changes per Treatment Group

There was good to excellent interrater agreement regarding the perceived changes in facial profile appearance overall, as well as per treatment group (ICC > 0.85, Table 2, Figure 4).

The variances of all dependent variables were equal across groups (Levene’s test, *p* > 0.01). Multivariate tests showed a small, although significant effect of rater type on the assessments of changes in facial profile appearance (F = 2.17, *p* = 0.007, Pillai’s Trace = 0.22, partial η^2^ = 0.07). On the contrary, there was a significant, substantial effect of treatment group on the assessments (F = 20.94, *p* < 0.001, Pillai’s Trace = 0.44, partial η^2^ = 0.44) and no combined effect of rater type and treatment group (F = 0.71, *p* = 0.774, Pillai’s Trace = 0.08, partial η^2^ = 0.03). Initial testing revealed that neither the sex factor nor its associated interactions had a statistically significant impact on the outcomes (*p* > 0.05). As a result, the sex factor was excluded from the multivariate model.

Individual ANOVA tests conducted for each dependent variable consistently yielded results that aligned with the findings from the multivariate analysis regarding treatment groups and the combined effects of treatment group and rater type. In these analyses the rater type factor was not significant in any case (Table 3 and Figure 5).

As illustrated in Figure 4 and Figure 5, and quantified in Table 4, apart from the upper lip, all rater groups consistently identified negligible positive changes in the appearance of facial profile after exclusively orthodontic treatment. In contrast, substantial positive changes were detected following combined orthodontic and orthognathic surgery interventions (*p* < 0.001).

Regarding the rater groups, there were few significant differences detected in post hoc analyses. These were between orthodontists and laypeople, with the later perceiving slightly less positive changes in the face (Diff.: −5.11, *p* = 0.048) and the lower lip than the former (Diff.: −6.45, *p* = 0.035; Table 5). Orthodontists also perceived slightly more positive changes on the lower lip than patients (Diff.: 6.21, *p* = 0.043). Overall, orthodontists tended to rate changes more positively, followed by surgeons, patients, and laypeople (Table 5).

## 4. Discussion

The present investigation used actual patient facial profile images to assess the changes in facial appearance induced by orthodontic camouflage versus orthognathic surgery interventions, as perceived by different rater groups. All rater groups identified minor positive changes in the facial profile appearance after exclusively orthodontic treatment in contrast to substantial positive changes detected following combined orthodontic and orthognathic surgery. We assume that comparable improvements were evident in all facial regions following surgery because of the multitude of factors involved in appearance perception, especially when presenting entire faces to the raters. Changes in the position/morphology of one facial structure might have affected the appearance/perception of neighbouring structures as well. The differences between the two treatment approaches tended to be slightly larger in the lower face and the chin as compared to the lips. The treatment effect on the overall facial profile appearance was similar to that on the lower face, underscoring a significant influence of the lower face on the overall profile. Earlier research with similar methodology focusing on adolescent patients with convex profiles who received orthodontic treatment for growth modification indicated 4–10% more positive changes in facial profile appearance compared to normal profile controls [4]. In the present study, the differences between the two interventions ranged from 14% to 18%, indicating a substantial impact of orthognathic surgery in the facial profile appearance.

The few previous studies that used actual patient images to compare the effects of orthodontic camouflage with surgical-orthodontic treatment presented conflicting results [19,21,24]. They used panels of judges, which rated randomized pre- and post-treatment facial photos for their attractiveness. Both studies did not differentiate between distinct rater types, whereas the study of Proffit et al. (1992) [21] included only specialists. Moreover, the study of Shell and Woods (2003) [19] compared adolescent patients that were treated through growth modification with adult patients that were subjected to orthognathic surgery. The latter study did not identify any differences between the two treatment approaches, whereas the previous one did identify an average improvement of about 5% in facial attractiveness following surgery, compared to no changes with orthodontic treatment alone. Both studies presented simultaneous projections of frontal and profile facial photos to raters, and thus, their outcomes are not directly comparable to ours [5,28,29], which concerned profile assessments.

The assessment of treatment outcomes by different groups related either to treatment decisions or to the actual impact of an intervention on patients’ lives defines the actual treatment efficiency. These groups are definitely the patients and the treating doctors, but also the laypeople, which represent the response of the society at a given outcome. The latter group is often ignored, despite its major importance, since facial appearance has a significant impact in various aspects of life, including personal, professional, and social life [3]. Thus, the outcome as perceived laypeople determines if the patients will actually receive the anticipated benefits from a given intervention. Previous studies have shown that different assessors may perceive treatment outcomes in different ways [27,30,31].

In this study, we included all relevant rater groups and investigated their agreement. The perceived changes in facial profile appearance overall, as well as per treatment group, were judged similarly by the rater groups. There were only small differences between rater groups, and these were statistically significant in few cases. Overall, orthodontists tended to rate changes more positively, followed by surgeons, patients, and laypeople. The more optimistic view of specialists as compared to non-specialists has been also shown previously [27,30,32], emphasizing the need for including non-specialist groups in analogous investigations.

The substantial positive effects of orthognathic surgery on facial profile appearance need to be considered in treatment decision making, especially in patients that seek to improve their facial appearance. The high impact of facial esthetics on various life aspects, as well as their pivotal role in motivating individuals to seek treatment and in patient satisfaction, designate the important clinical implications of the present findings. Treating doctors should consider them when consulting adult patients with convex facial profile and increased overjet. However, decisions must always be made in a cost–benefit manner, considering a multitude of factors that can vary significantly on an individual basis. Orthognathic surgery is an invasive procedure related to major and minor problems and complications, including postoperative malocclusion, inferior alveolar nerve injury, and bad split [33], and might lead in certain cases to unfavorable outcomes [14,17,20,21]. In such cases, orthognathic camouflage procedures may be indicated, instead of conventional surgery, as a viable, less invasive option [34]. In selected cases, this option could also be considered in combination with orthodontic camouflage procedures that enhance dental esthetics, to attain favorable facial outcomes, while minimizing the risks and the burden of treatment.

In agreement with previous studies, camouflage orthodontic treatment did not affect the facial appearance of convex profile patients [21]. This does not mean that the patients receive no benefit from orthodontic treatment in terms of appearance. Dental and smile esthetics might still be considerably affected in a positive way [35], although this remains to be tested. On the contrary, the impact of orthognathic surgery on facial appearance was very high, considering the variety of factors involved [3,36,37] and that this intervention alters only facial morphology. Under this prism, an impact of almost 20 points towards the positive side of the facial appearance scale is considered substantial.

In terms of treatment outcome retention, recent evidence in adolescents indicated that the stability of Class II correction with functional appliance therapy and orthodontic camouflage after a 12-month retention period was acceptable and did not differ between the two groups [38]. To our knowledge, there is no similar study including orthodontic camouflage interventions in adult convex profile patients, which might present increased post-treatment overjet, with no anterior dental contacts. An important long-term risk in these cases could be the overeruption of lower incisors leading to contact with the palate during intercuspation. This could cause severe periodontal problems at the upper incisors [39]. To avoid this risk, a life-long Hawley type retainer in the maxilla, with anterior contacts at the lower incisors or a lower fixed retainer, with its distal end teeth having occlusal contacts with their antagonists, should be implemented.

The present study employed slightly modified actual facial profile photos, maintaining equal shape and size while removing prominent marks. This approach aimed to present realistic, real-life images to the raters, including non-specialists who might not be familiar with facial profile outlines [40,41]. This allowed for a holistic assessment of the changes in facial appearance, which involves factors that affect esthetic perception, such as the eye colour, the skin texture, or the colour of hair [36,37]. According to a previously published protocol [4,5], the raters were asked to assess the changes between two facial photos of the same individual, presented simultaneously. This way the raters assessed realistic patients’ images, while factors other than facial morphology that could affect ratings were controlled. Profile photos were presented since this is the main area where all interventions target [4,42], and, thus, treatment induced changes might be better detectable. On the other hand, previous studies indicated that judgments may by modified when other facial aspects are also presented, which is considered a more representative approximation of real life conditions [5,19,27,32]. Future research should present simultaneously additional facial aspects to the raters to test if the differences between the interventions will be affected. There were no statistically significant differences between the treatment groups before treatment and the groups had comparable overjet after treatment. Orthognathic surgery patients demonstrated greater improvement in the facial contour angle, a reliable indicator of the study outcome [43]. These findings validate the suitability of the current sample for exploring the study hypothesis and its reflection of real-world clinical scenarios. However, a tendency towards slightly greater facial convexity and overjet in the surgical cases before treatment should also be noted. Finally, apart from the inclusion criteria, the selection of the orthognathic patients was carried out irrespective of the applied surgical approach (e.g., single or double jaw surgery), which might differ among clinicians even in similar clinical conditions. It was assumed that the surgical plan was designed to achieve reasonable treatment goals in terms of improvement of facial appearance, which were defined by the patient and treating doctor on an individual basis. For this reason and also due to sample size considerations, we refrained from reporting more detailed information regarding specific surgical as well as orthodontic treatment plans applied in the sample.

## 5. Conclusions

The results of the study reject the hypothesis that the perceived changes in facial profile appearance are not affected by the type of intervention. According to diverse groups, including laypeople, the combined orthodontic and orthognathic surgery interventions were efficient in improving the facial appearance of patients with a convex profile. On the other hand, orthodontic treatment alone did not have significant effects on patients’ facial appearance.

Before initiating treatment, healthcare providers should communicate to non-growing convex profile patients that, despite the surgical risks associated with orthognathic surgery, the improvement in profile appearance is significantly greater with the surgical approach. This is particularly important given the pivotal role of facial esthetics in the decision-making process.

## Figures and Tables

**Figure 2 jcm-13-00091-f002:**
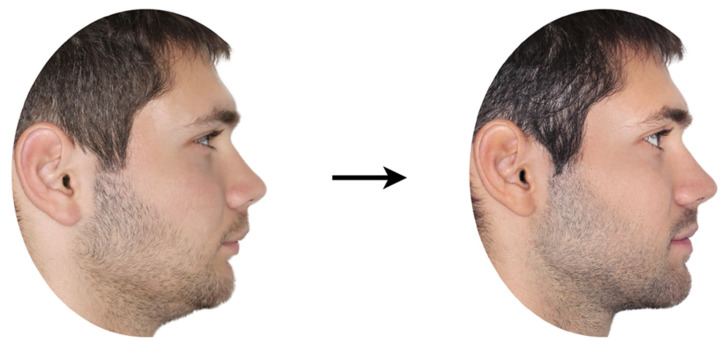
Pre- (**left**) and post-treatment (**right**) patient images, as presented to the raters.

**Figure 3 jcm-13-00091-f003:**
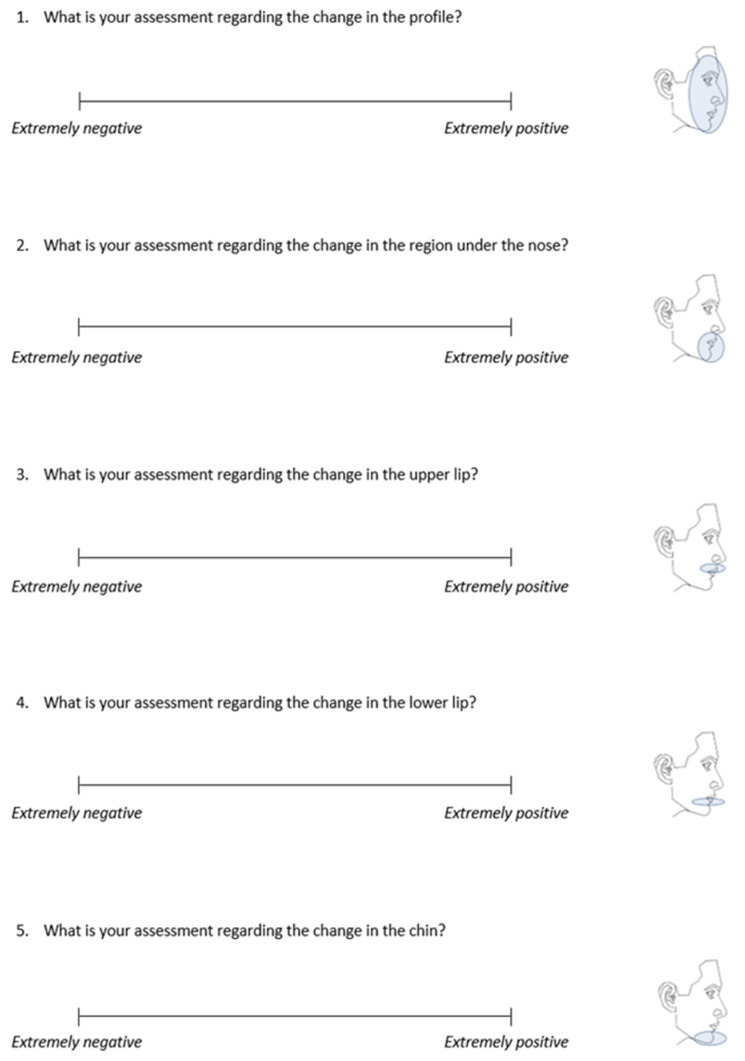
Questionnaire used for the assessment of changes in facial profile appearance from pre- to post-treatment through the visual analogue scale (VAS).

**Figure 4 jcm-13-00091-f004:**
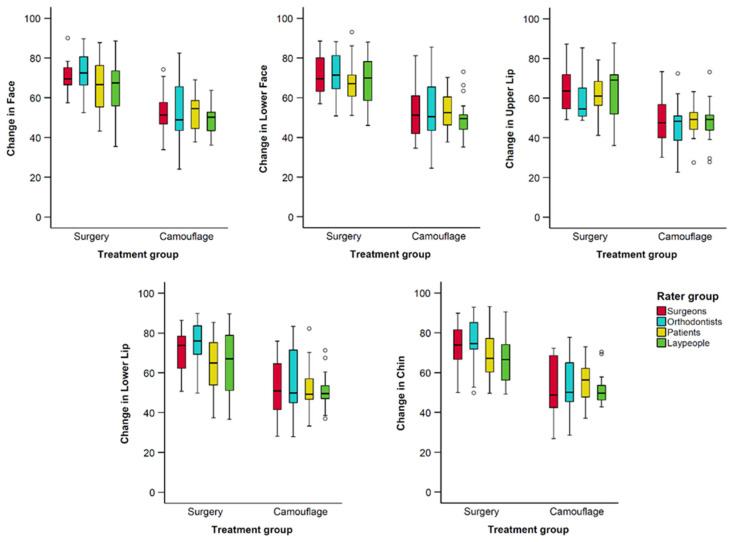
Box plots showing the assessed changes from pre- to post-treatment condition in VAS values (y-axis), grouped by rater type. The upper limit of the black line represents the maximum value, the lower limit the minimum value, the boxed the interquartile range, and the horizontal black line the median value. Outliers (>±3 SD) are shown as black circles.

**Figure 5 jcm-13-00091-f005:**
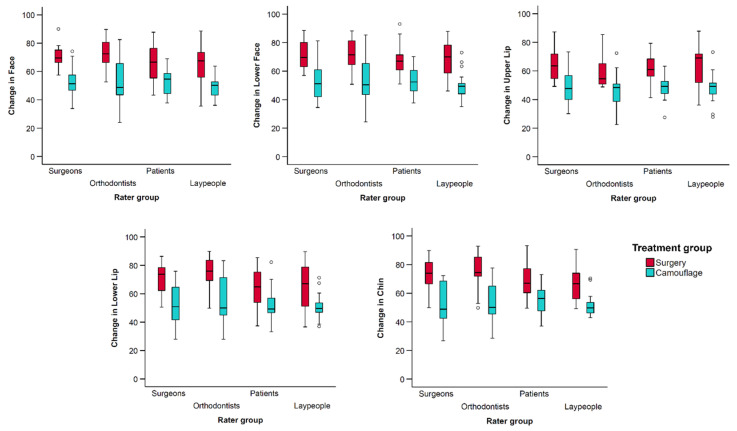
Box plots showing the assessed changes from pre- to post-treatment condition in VAS values (y-axis), grouped by treatment approach. The upper limit of the black line represents the maximum value, the lower limit the minimum value, the boxed the interquartile range, and the horizontal black line the median value. Outliers (>±3 SD) are shown as black circles.

**Table 1 jcm-13-00091-t001:** Overview of the patient sample characteristics.

Treatment Type	N (Sex)	Age (yrs) Mean ± SD		Treatment Duration (yrs) Mean ± SD	Facial Contour Angle (°) Mean ± SD			Overjet (mm) Mean ± SD	
		T0	T1		T0	T1	T1-T0	T0	T1
Camouflage	18 (8 M, 10 F)	22.7 ± 8.3	25.1 ± 8.1	2.5 ± 0.8	20.2 ± 3.3	19.0 ± 3.5	−1.2 ± 2.1	7.6 ± 2.2	3.9 ± 1.6
Surgery	18 (8 M, 10 F)	23.9 ± 7.4	27.1 ± 7.1	3.1 ± 1.3	22.3 ± 6.6	16.2 ± 6.3	−6.2 ± 3.9	8.9 ± 2.8	3.6 ± 1.9
*p*-value *	-	0.643	0.442	0.078	0.169	0.189	<0.001 *	0.137	0.629

M: males, F: females, yrs: years, T0: pre-treatment, T1: post-treatment; * Bonferroni adjusted level of significance: *p* < 0.01.

**Table 2 jcm-13-00091-t002:** Interrater agreement of VAS ratings among groups of raters, tested through the intraclass correlation coefficient (ICC; two-way mixed model, absolute agreement, results regarding average measures; 95% confidence intervals reported in the parentheses).

	Face	Lower Face	Upper Lip	Lower Lip	Chin
Total	0.93 (0.89, 0.96)	0.95 (0.92, 0.97)	0.94 (0.89, 0.96)	0.94 (0.89, 0.97)	0.92 (0.87, 0.96)
Camouflage	0.88 (0.76, 0.95)	0.92 (0.85, 0.97)	0.92 (0.83, 0.97)	0.92 (0.83, 0.96)	0.85 (0.70, 0.94)
Surgery	0.87 (0.73, 0.95)	0.91 (0.82, 0.96)	0.90 (0.79, 0.96)	0.91 (0.79, 0.97)	0.87 (0.73, 0.95)

**Table 3 jcm-13-00091-t003:** Results of the ANOVAS testing the effect of rater type and treatment group on the assessed changes from pre- to post-treatment condition.

Source	Dependent Variable	df	F	Sig.
Treatment group	Face ^a^	1	93.61	<0.001
	Lower Face ^b^	1	83.19	<0.001
	Upper Lip ^c^	1	53.06	<0.001
	Lower Lip ^d^	1	54.19	<0.001
	Chin ^e^	1	82.99	<0.001
Rater type	Face	3	1.48	0.222
	Lower Face	3	0.61	0.611
	Upper Lip	3	0.64	0.588
	Lower Lip	3	2.00	0.117
	Chin	3	0.96	0.415
Treatment group × Rater type	Face	3	0.53	0.661
	Lower Face	3	0.45	0.715
	Upper Lip	3	0.15	0.929
	Lower Lip	3	0.54	0.658
	Chin	3	1.22	0.306

^a^ R Squared = 0.42 (Adjusted R Squared = 0.39), ^b^ R Squared = 0.39 (Adjusted R Squared = 0.36), ^c^ R Squared = 0.29 (Adjusted R Squared = 0.25), ^d^ R Squared = 0.31 (Adjusted R Squared = 0.28), ^e^ R Squared = 0.40 (Adjusted R Squared = 0.37). df: degrees of freedom. F: F-value. Sig.: Significance shown as *p*-values.

**Table 4 jcm-13-00091-t004:** Estimated marginal means per treatment group and associated pairwise comparisons.

Dependent Variable	Treatment Group	Mean	Std. Error	95% Confidence Interval		Mean Difference (Surgery—Camouflage)	Std. Error	Sig.
				Lower Bound	Upper Bound			
Face	Camouflage	51.56	1.28	49.03	54.09	17.50	1.81	<0.001
	Surgery	69.07	1.28	66.54	71.60			
Lower Face	Camouflage	52.34	1.36	49.64	55.03	17.60	1.93	<0.001
	Surgery	69.93	1.36	67.24	72.63			
Upper lip	Camouflage	48.30	1.38	45.59	51.02	14.16	1.94	<0.001
	Surgery	62.46	1.38	59.74	65.18			
Lower lip	Camouflage	52.72	1.52	49.72	55.72	15.78	2.14	<0.001
	Surgery	68.50	1.52	65.50	71.50			
Chin	Camouflage	52.85	1.40	50.09	55.61	17.97	1.97	<0.001
	Surgery	70.82	1.40	68.06	73.57			

Sig.: Significance shown as *p*-values.

**Table 5 jcm-13-00091-t005:** Pairwise comparisons per rater group based on estimated marginal means.

Dependent Variable	(I) Rater	(J) Rater	Mean Difference (I − J)	Std. Error	Sig.	95% Confidence Interval for Difference	
						Lower Bound	Upper Bound
Face	Laypeople	Surgeons	−3.37	2.56	0.190	−8.43	1.69
	Laypeople	Orthodontists	−5.11	2.56	0.048 *	−10.17	−0.05
	Laypeople	Patients	−1.64	2.56	0.523	−6.70	3.42
	Surgeons	Orthodontists	−1.74	2.56	0.498	−6.80	3.32
	Surgeons	Patients	1.73	2.56	0.500	−3.33	6.79
	Orthodontists	Patients	3.47	2.56	0.177	−1.59	8.53
Lower Face	Laypeople	Surgeons	−2.73	2.73	0.319	−8.13	2.67
	Laypeople	Orthodontists	−3.23	2.73	0.239	−8.63	2.17
	Laypeople	Patients	−0.99	2.73	0.718	−6.39	4.41
	Surgeons	Orthodontists	−0.50	2.73	0.855	−5.90	4.90
	Surgeons	Patients	1.74	2.73	0.524	−3.65	7.14
	Orthodontists	Patients	2.24	2.73	0.413	−3.16	7.64
Upper Lip	Laypeople	Surgeons	−0.18	2.75	0.949	−5.61	5.26
	Laypeople	Orthodontists	3.19	2.75	0.249	−2.25	8.62
	Laypeople	Patients	1.43	2.75	0.604	−4.01	6.87
	Surgeons	Orthodontists	3.36	2.75	0.224	−2.08	8.80
	Surgeons	Patients	1.61	2.75	0.560	−3.83	7.04
	Orthodontists	Patients	−1.76	2.75	0.524	−7.19	3.68
Lower Lip	Laypeople	Surgeons	−3.30	3.03	0.279	−9.29	2.70
	Laypeople	Orthodontists	−6.45	3.03	0.035 *	−12.44	−0.45
	Laypeople	Patients	−0.24	3.03	0.937	−6.24	5.76
	Surgeons	Orthodontists	−3.15	3.03	0.301	−9.15	2.85
	Surgeons	Patients	3.06	3.03	0.315	−2.94	9.05
	Orthodontists	Patients	6.21	3.03	0.043 *	0.21	12.20
Chin	Laypeople	Surgeons	−2.99	2.79	0.286	−8.51	2.53
	Laypeople	Orthodontists	−4.66	2.79	0.097	−10.18	0.85
	Laypeople	Patients	−2.68	2.79	0.338	−8.20	2.84
	Surgeons	Orthodontists	−1.68	2.79	0.549	−7.19	3.84
	Surgeons	Patients	0.31	2.79	0.912	−5.21	5.83
	Orthodontists	Patients	1.98	2.79	0.478	−3.532	7.50

* The mean difference is significant at the 0.05 level.

## Data Availability

All data are available in the main text. The protocols and the anonymized datasets generated and/or analyzed during the current study are available from the corresponding author on reasonable request. The patient’s photos and any other identifying information cannot be shared.

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
