# Peer review of "Perceived Effects of Orthognathic Surgery versus Orthodontic Camouflage Treatment of Convex Facial Profile Patients"

_jcm, 2023, doi:10.3390/jcm13010091_

Round 1

Reviewer 1 Report

Comments and Suggestions for Authors

line 43: I would suggest to write that profile is one main reason to seek for treatment. (Another main reason is bad aesthetics of the teeth. Therefore, patients often visit an orthodontic clinic first). see line 347 in the discussion!

line 56: The amount of - please delete one blank space before "of"

line 131: In my opinion, it would have been better to take the profile photographs in the centric condylar position. Here the profile is often even more impaired than in the maximum intercuspidation.

Line 146: Sorry, but I do not understand why you write, that ten members of each group rated the patients’ pictures while in Line 151 thirty members of each group rated the picutres. In Line 208 you wrote again, that ten members per group and 20 laypeople rated the pictures.

line 217: I think you can leave out "the" before orthognatic surgery and conventional

line 110 and line 234: With regard to the difference between operated and non-operated patients, I would recommend to include a sketch of the facial contour angle in the paper for explanation, as in my view this angle does not represent a generally accepted standard like the values of the cephalometric radiograph

Table 1: Overjet of Camouflage-Patients was 7.6± 2.2. In line 106 you wrote that the overjet hat to be between 6 and 12 mm. could it be that the overjet was allowed to be smaller than 6?

Table 2: Sorry, but I think that the title is misleading. There is no breakdown by rater group here and the values in brackets should also be clearly explained. Are these confidence intervals?

Line 351: this sentence seems very misleading to me...

Discussion:

Unfortunately, there is no distinction between monognathic and bignathic surgery in the surgical collective. The possibilities for improving the profile are much greater with the freedom of bimaxillary surgery than by simply advancing the mandible. This should at least be discussed. Patients often choose a bimaxillary procedure precisely to improve the profile. It should also be pointed out that there are also objective methods of evaluating the facial profile. For example, the Rob Mulié /Rinko Brons method can be used. In my opinion, it should also be discussed that the profile improvement in all facial regions was better in the nearly same amount if surgery was used. Actually, one would expect the change on the chin and lower lip to be much more pronounced than on the entire face and especially on the upper lip. Is there an explanation for this?

Conclusion:

I think it should be made clear that the study clearly refutes the hypothesis formulated in the introduction. As expected, the profile improvement was significantly better with surgery, and this in all regions of the face. I think that this fact should also be explained to all patients before starting treatment, because many patients then decide not only for surgery, but also for bimaxillary surgery to improve facial aesthetics. Lines 392 to 395 should be included in the discussion and can be deleted from the conclusion.

Reviewer 2 Report

Comments and Suggestions for Authors

The title of the evaluated manuscript: Perceived effects of orthognathic surgery versus orthodontic 2 camouflage treatment of convex facial profile patients.

This manuscript evaluated the outcome of the orthodontic/orthognathic vs camouflage orthodontic treatment for 36 patients with convex profile (Angle class II division I), based on before-after sets of photos shown to both professionals and patients/laypeople. The results seem to suggest that the orthodontic/orthognathic approach might produce better aesthetic results than only orthodontic approach.

Introduction: In my opinion a more detailed description of the advantages and limits of the orthodontic/orthognathic and camouflage approaches would enforce the manuscript. A practitioner reader must have all the information for a better understanding.

Methods:  more figures of the before and after treatment cases should be introduced. Fig1, display small changes especially on mandibula, thus, more pictures for emphasizing the two therapeutical approaches is needed.

Discussion: orthognathic-orthodontic approach is an extremely invasive treatment, with immediate and medium-long term risks. Thus, in my opinion a more info regarding the risk-benefit of the two approaches must be added. Moreover, the 14-18% satisfaction after the orthodontic-orthognathic approach when compared with orthodontic treatment must be sustained by the gravity of the prognosis (showed by the displayed photos). In fig 1, there as only small changes, thus, more before-after pictures should be added to justify the need for such an invasive approach.

Conclusions: the significant impact changes displayed by the surgical approach must be shown in the manuscript by before-after pictures. Moreover, the high risks of the surgical approach must be balanced by the significant results of the after-treatment aesthetics. I did not see this aspect, thus in my opinion this aspect must be addressed during the review phase.  

Round 2

Reviewer 2 Report

Comments and Suggestions for Authors

Accept as it is